# The Relationships between Damaging Behaviours and Health in Laying Hens

**DOI:** 10.3390/ani12080986

**Published:** 2022-04-11

**Authors:** Virginie Michel, Jutta Berk, Nadya Bozakova, Jerine van der Eijk, Inma Estevez, Teodora Mircheva, Renata Relic, T. Bas Rodenburg, Evangelia N. Sossidou, Maryse Guinebretière

**Affiliations:** 1Direction de la Stratégie et des Programmes, French Agency for Food, Environmental and Occupational Health & Safety (ANSES), 94701 Maisons-Alfort, France; 2Institute for Animal Welfare and Animal Husbandry, Friedrich-Loeffler-Institut, 29223 Celle, Germany; jutta.berk@tiho-hannover.de; 3Department of General Animal Breeding, Animal Hygiene, Ethology and Animal Protection Section, Faculty of Veterinary Medicine, Student’s Campus, Trakia University, 6000 Stara Zagora, Bulgaria; nadiab@abv.bg; 4Animal Health and Welfare, Wageningen Livestock Research, Wageningen University and Research, De Elst 1, 6708 Wageningen, The Netherlands; jerine.vandereijk@wur.nl; 5Department of Animal Production, Neiker-Basque Institute for Agricultural Research and Development, 01080 Vitoria-Gasteiz, Spain; iestevez@neiker.eus; 6Section of Biochemistry, Faculty of Veterinary Medicine, Trakia University, 6000 Stara Zagora, Bulgaria; teodoramirchevag@abv.bg; 7Faculty of agriculture, University of Belgrade, 11080 Belgrade, Serbia; rrelic@agrif.bg.ac.rs; 8Animals in Science and Society, Faculty of Veterinary Medicine, Utrecht University, Yalelaan 2, 3584 Utrecht, The Netherlands; t.b.rodenburg@uu.nl; 9Laboratory of Farm Animal Health and Welfare, Veterinary Research Institute, Ellinikos Georgikos Or-Ganismos-DIMITRA (ELGO-DIMITRA), 57001 Thessaloniki, Greece; sossidou@vri.gr; 10Epidemiology, Health and Welfare Unit, French Agency for Food, Environmental and Occupational Health & Safety (ANSES), 22440 Ploufragan, France; maryse.guinebretiere@anses.fr

**Keywords:** hen health, damaging behaviour, laying hens, housing system

## Abstract

**Simple Summary:**

The design of housing systems and genetic selection of laying hens have in the past focused mainly on productivity, excluding issues around the animals’ behavioural needs and welfare. Because of inadequate housing conditions and especially a barren environment, behavioural disorders such as feather and body pecking, as well as cannibalism, occur in the modern layer industry. Since conventional cages for egg production were banned in the European Union in January 2012, alternative systems such as floor, aviary, free-range, and organic systems have become increasingly common and now concern over 50% of hens housed in Europe. Despite the many advantages that come with non-cage systems, the shift to a housing system where laying hens are kept in larger groups and more complex environments has given rise to new challenges related to management, health, and welfare. We have carried out a review showing the close relationships between damaging behaviours and health in modern husbandry systems for laying hens.

**Abstract:**

Since the ban in January 2012 of conventional cages for egg production in the European Union (Council Directive 1999/74/EC), alternative systems such as floor, aviary, free-range, and organic systems have become increasingly common, reaching 50% of housing for hens in 2019. Despite the many advantages associated with non-cage systems, the shift to a housing system where laying hens are kept in larger groups and more complex environments has given rise to new challenges related to management, health, and welfare. This review examines the close relationships between damaging behaviours and health in modern husbandry systems for laying hens. These new housing conditions increase social interactions between animals. In cases of suboptimal rearing and/or housing and management conditions, damaging behaviour or infectious diseases are likely to spread to the whole flock. Additionally, health issues, and therefore stimulation of the immune system, may lead to the development of damaging behaviours, which in turn may result in impaired body conditions, leading to health and welfare issues. This raises the need to monitor both behaviour and health of laying hens in order to intervene as quickly as possible to preserve both the welfare and health of the animals.

## 1. From “Productivism” Systems to More “Welfare-Friendly” Approaches: New Challenges to Face

After the Second World War, animal production systems were automatised and rationalised in order to reach higher productivity to be able to feed Europe with cheap animal protein. The spectacular development of poultry husbandry systems until the 1990s led to systems that were optimal in terms of working conditions, productivity, and food safety and animal health, for example, with the separation of animals and eggs from manure, such as in cages systems. Housing system design and the genetic selection of animals focused on productivity, excluding considerations around animal behavioural needs and welfare [1,2]. Due to inadequate housing conditions, and especially the barren environment, behavioural disorders such as feather pecking, toe pecking, vent/cloacal pecking, and cannibalism can occur [3].

Since the ban in January 2012 of battery cages for egg production in the European Union (Council Directive 1999/74/EC), alternative systems such as floor, aviary, free-range, and organic systems have become increasingly common, reaching 50% of hen housing in Europe in 2019 [4]. Non-cage (alternative) systems provide the birds much more behavioural freedom as well as ample access to litter, nests, and perches, which improves their welfare. Additionally, free-range and aviary systems allow higher bird activity, which may result in increased bone density and strength [5]. For instance, free-ranging laying hens have been shown to have better plumage conditions and higher final body weights, which leads to higher egg weight than hens in indoor systems [5,6,7].

Despite the many advantages associated with non-cage systems, the shift to a housing system where laying hens are kept in larger groups and more complex environments has given rise to new challenges related to management, health, and welfare. Considerable research has been performed to study environmental conditions and management practices in non-cage systems in different climatic conditions [7,8,9,10]. For instance, several studies have found that aviaries can have a negative impact on indoor air quality, with higher concentrations of suspended dust than in cage systems, resulting from the presence of floor litter (higher ammonia levels) and hens’ activities (higher particulate matter levels) in it [11]. Dust is composed of inorganic and organic compounds from the birds themselves as well as from feed, litter, and building materials [12]. Dust may be a vector of microorganisms and toxins. High dust levels may compromise the health and welfare of both birds and their caretakers [13,14]. Bird health can also be negatively affected in non-cage systems by a higher risk of bacterial and fungal infections spreading among the birds [15,16].

Finally, welfare challenges persist, even after the switch to non-cage systems, including keel bone damage (reviewed by Riber et al. [17]), feather pecking, toe pecking, vent/cloacal pecking, and cannibalism [16,18]. Damaging pecking may occur during rearing periods of pullets as well as during the laying period [19,20,21], even though it is more prevalent in the laying period. Severe feather pecking, leading to feather loss, can result in economic losses as a result of increased food consumption in defeathered birds [22,23] and increased mortality [24,25], as well as in reduced animal welfare since feather pecking is painful for the birds being pecked [26]. Additionally, hens with feather damage are more susceptible to cannibalistic pecking [27]. Free-range systems are also associated with a higher risk of exposure to parasites, pathogens, and predation [28].

## 2. Description and Definition of the Different Concepts Linked to Health and Damaging Behaviours

### 2.1. Health of Laying Hen Flocks

There are many ways to define the health of productive animals. Considering the World Health Organisation definitions of 1946 and 2006, health is “a state of complete physical, mental and social well-being and not just the absence of disease or infirmity”. Animal health can be defined as “a lack of disease or normal functioning of the animal body and normal behaviour” [28]. Here, we see that the development of abnormal behaviour is considered an impairment of animal health. In the production sector, Gunnarson [29] defines health as “the state of the animal organism that allows highest productivity based on a balance between animals and their environment, as well as the animal’s physical well-being”. More recently, animal health has been considered one of the pillars of the “One Health” concept, developed with the aim of protecting public health [30]. The vision of One Health is that human health can be better protected through policies that ensure the health of animals and of ecosystems since human, animal, and environmental health are all interconnected [30]. Within the One Health framework, animal welfare offers opportunities to define the conditions for animals to grow healthily and to be able to cope with pathogens while reducing the need for the use of antibiotics. Such conditions are defined by the animals´ behavioural needs that have been shaped by their own evolutionary history and are deeply imbedded in their genetic makeup. Understanding the factors that affect the social behaviour of laying hens [31,32,33], or their responses to the features of their surrounding environment [6,34,35], provides the scientific information needed to manage flocks according to these biological needs, to avoid sources of potential stressors, and to reduce the risk of damaging behaviour. This type of holistic approach will help to preserve animal health and welfare while allowing optimal animal performance in modern animal production systems. From the definitions of health cited above, we can see that they include the mental state of an animal and that both physical and mental health can be captured in the term “welfare”. The French Agency for Food, Environmental and Occupational Health & Safety (ANSES) [36] defines the welfare of an animal as “its positive mental and physical state as related to the fulfilment of its physiological and behavioural needs, in addition to its expectations”. Importantly, when the behavioural needs or expectations of animals are not fulfilled, damaging behaviours may develop or increase. The next chapter discusses the most common damaging behaviours that can be encountered in current intensive housing systems for layers and that may compromise animal welfare.

### 2.2. Damaging Behaviours in Laying Hens

#### 2.2.1. Pecking Behaviours in Chickens

Pecking is a natural behaviour in chickens during foraging and exploration of the environment. When a chicken pecks a conspecific, a distinction is made between pecking arising from aggressive or non-aggressive motivations, as the body parts targeted and risk factors for the behaviour differ. Non-aggressive injurious pecking is considered a redirected form of foraging behaviour, as both pecking during feeding and injurious pecking show similar fixed motor patterns [37]. An association has been found between the high occurrence of litter-directed pecks by individuals when they are young and a high level of severe feather pecking and litter-directed pecks when they are adults [37]. This suggests that severe feather pecking is not a direct substitute for foraging but that some individuals have high pecking motivation overall and are, thereby, more prone to develop injurious pecking in addition to foraging.

In cage-free systems, hens have greater behavioural opportunities and freedom of movement, but these systems may also be associated with a greater risk of damaging behaviours as compared to cages [20,38]. Even though these behaviours can still happen in cages, they are limited to the cage where they develop.

#### 2.2.2. From Pecking Behaviour to Damaging Behaviour

Damaging behaviours represent a collection of unwanted behaviours that develop under certain circumstances at high frequency and intensity in laying hens, other poultry species, and avian species [39] and can cause harm to other group members. They include feather pecking [40,41,42], aggressive pecking [43] (outside of the frame of hierarchy establishment), different forms of cannibalism [44,45,46], which include vent/cloacal pecking [42,47], and toe pecking [48,49].

Gentle feather pecking is a frequent behaviour in young birds and is also important in social recognition. In adult birds, stereotyped gentle feather pecking can be observed, where birds, for instance, spend a long time pecking at the tips of the tail feathers of another bird. Although this behaviour indicates a welfare problem for the pecker, it usually does not lead to much feather damage [45,50]. The main problematic behaviour is severe feather pecking, directly affecting the health of the hens—several feathers are lost, or whole stripping of certain areas of the body is observed. This is associated with pain in the affected hen [26] and can cause skin eruption or bleeding. Severe cases of feather pecking can escalate into cannibalism and death of the bird victim, cannibalistic tissue pecking, and vent/cloacal pecking, potentially leading to severely wounded or dead birds [45,46,50].

Cannibalistic behaviour involves beak-inflicted damage followed by the consumption of blood and tissues of conspecifics while they are still alive or after death [51]. Cannibalistic behaviour is learned by individual birds and can spread to others through social learning [51], even through adjacent cages [52]. Severe feather pecking can lead to increased risks of cannibalism [53]. Cannibalism in the cloacal area, also known as “vent pecking”, is considered a distinct form of cannibalistic pecking [42] and may negatively affect the welfare and health of the bird by causing considerable pain and even leading to mortality [54]. Serious inflammatory and even infectious processes can follow skin breakage. Toe pecking is another behaviour that is harmful to victims and that negatively affects hen health. It occurs when a bird starts to peck the toes of another bird [48,49]. In severe forms, toe swelling can be attributed to cannibalism, and complications may be lethal [55].

#### 2.2.3. Main Causes of Damaging Behaviour and Control Strategies

Regarding the causal factors leading to feather pecking, a classical hypothesis suggests that it is a redirected form of foraging that develops in the absence of foraging material [56,57,58,59]. The hypothesis is that under commercial conditions where chicks are reared in the absence of their mother’s guidance, the direction of foraging pecks toward flock mates could result from a chick’s failure to learn to direct these pecks toward appropriate substrates and food items. In addition, the absence of suitable manipulable foraging material can lead to injurious pecking in chicks [37]. In a review, De Haas et al. [37] explored how behavioural programming via prenatal conditions (role of maternal stress, egg conditions, incubation settings) and early postnatal conditions (chick brooding conditions) could influence the development of injurious pecking in laying hens. This review argues that it may be possible to prevent injurious pecking in commercial laying hen flocks by adapting the environmental conditions of previous generations, optimising incubation conditions, reducing stress around hatching, and guiding the early learning of chicks.

Damaging behaviour can emerge at different ages in most breeds, although with varying intensity depending on the genetic line [34], and can affect a large number of birds in the flock. Reported percentages of affected flocks at the end of lay can reach values as high as 60% of the flocks, with more than 10% of hens having moderate or severe feather damage in one body region [21], or 86% of the flocks in which severe feather pecking was observed [60].

Although no strategy can guarantee the complete absence of pecking behaviours, optimised management practices, especially concerning feeding, lighting, and climatic conditions [35] and environmental enrichment in pullets and adult birds [61,62,63], can help to reduce the risk. Access to outdoor free-range areas is associated with plumage preservation [6,7,64] and a reduced risk of injurious pecking [65]. Genetic selection at the commercial scale will help in the control of feather pecking [41,44,66]. For instance, Rodenburg et al. [67] offer various genetic means to limit feather pecking, cannibalism, and vent/cloacal pecking based on the systematic selection of birds with less-pronounced damaging behaviours than other birds.

Another hypothesis suggests that mild feather pecking could be a redirected form of social grooming and may have a social recognition function [68]. Kjaer et al. [69,70] suggest that severe feather pecking is related to neurological changes that cause hyperactivity, although Krause et al. showed that selection for high locomotor activity did not result in an increase in feather pecking [71]. Recent studies found that genes involved in cholinergic signalling, channel activity, synaptic transmission, and immune response are involved in feather-pecking mechanisms [66].

Although Borda-Molina et al. did not find any relations between microbiota and feather pecking [72], there is growing evidence that gut microbiota influence hens behaviour and physiology [73,74]. However, whether microbiota can influence the development of feather pecking is not fully demonstrated [74,75]. This shows the complexity of the situation, involving the modulation by the gut microbes of the immune system, or maybe brain function not modulated through the immune system.

The way neurophysiology, gut physiology, and health in a broader sense impact the development of damaging behaviours in layers is described in the chapter below, immediately followed by the description of how, in return, the consequences of damaging behaviours will impact the health and welfare of animals.

## 3. Inter-Relationships between Damaging Behaviours and Health Problems in Current Housing Systems for Layers

Some damaging behavioural patterns may be associated with certain diseases in hens.

### 3.1. Recent Knowledge about the Impact of the Health Condition, Including Immune Status of Animals, on the Occurrence of Damaging Behaviours

#### 3.1.1. Immune System

The immune system plays a critical role in brain development. In particular, microglia (macrophage-like immune cells in the brain) have been shown to be involved in many aspects of brain development, such as synapse formation and neuronal survival [76]. Cytokines, chemokines, major histocompatibility complex (MHC) molecules, and toll-like receptors (TLRs) have been shown to play a critical role in neural development [76,77,78]. Cytokines can target neurocircuits that are involved in regulating mood, motor activity, motivation, and anxiety [79]. As a result, the immune system could influence behaviours through its role in brain development.

The immune system is also more directly involved in regulating behaviour. Cytokines and chemokines can alter behaviour, for example, in sickness behaviour, where sick animals show reduced feed and water intake, lower activity levels, decreased exploration and social interactions, and increased sleep [80,81]. Cytokines could influence behaviour via their effects on the synthesis, re-uptake, and release of neurotransmitters, such as serotonin, dopamine, and glutamate [79,82,83]. As an example, cytokines can influence the functioning of the hypothalamic-pituitary-adrenal axis (HPA axis). They can activate corticotropin-releasing hormone (CRH) and thereby stimulate the release of adrenocorticotropic hormone (ACTH) or can stimulate ACTH release, directly resulting in glucocorticoid release [79,84]. Cytokines can, in fact, influence behaviour via multiple routes.

In humans, there are similarities between sickness behaviour and behaviour expressed by individuals with certain neuropsychiatric disorders, such as depression [85]. Furthermore, many psychiatric disorders have been linked to immune dysregulation, including schizophrenia, anxiety and stress disorders, autism, and major depressive disorder [77].

Several studies have found relationships between the immune system and feather pecking. Most show genetic associations between feather damage (as an indicator of feather pecking) and the immune system. As mentioned previously, cytokines can influence the serotonergic and dopaminergic systems, and, in turn, these systems seem to be involved in the development of damaging behaviours such as feather pecking (for a review, see de Haas and van der Eijk [86]). In addition, through their effects on HPA axis functioning, cytokines could further influence how animals respond to or cope with stress. Feather pecking has been linked to coping styles and increased stress sensitivity [50,87]. Furthermore, feather pecking has been linked to motor activity [70], motivation, and fearfulness [88], and cytokines target brain areas that are involved in the regulation of these behaviours. The serotonergic and dopaminergic systems also appear to be dysregulated in many of these brain areas when feather pecking occurs [86]. The immune system may, therefore, play a role in the development of feather pecking.

Genetic associations have been found between immune-related genes, such as interleukin (IL4, IL9), nuclear factor NF-kappa-B (NFKB), chemokine (CCL4) genes, and feather damage score, providing evidence of a relationship between feather pecking and immunity at the genetic level [89]. Genetic mutations in the *IL4* and *IL9* genes were also associated with levels of natural antibodies (NAb) IgM and IgG [90]. NAb are antibodies that can bind antigens without prior exposure to the antigen [91]. These associations were mostly associative genetic effects on feather damage scores and not direct genetic effects, suggesting that NAb levels may be related to the propensity to perform feather pecking. This is further supported by the finding that when cage mates had higher NAb IgG levels, the individual had more feather damage [90]. Genetic associations were further found between severe feather pecking and specific antibody responses [92], indicating that there are genes simultaneously involved in both feather pecking and specific antibody response. Interestingly, several genes involved in immune responses, for example, TNF ligand and mitogen-activated protein kinase, were either upregulated or downregulated in the hypothalamus of feather-pecking birds compared to neutrals and victim birds [40]. Furthermore, a chicken line performing more feather damage showed upregulation of genes related to immune system processes in the brain compared to a chicken line showing less feather damage [93,94]. These findings provide additional arguments supporting a relationship between the immune system and feather pecking (see also Brunberg et al. [95]).

Further evidence for a relationship between the immune system and feather pecking comes from lines that were divergently selected on feather pecking and that differ in several immune parameters. High-feather-pecking (HFP) birds showed a higher antibody response to infectious bursal disease virus vaccination, while low-feather-pecking (LFP) birds had a higher number of white blood cells and higher expression of MHC class I molecules on T (CD4, CD8) and B cells [96]. Recently, the FP selection lines were shown to differ in both innate and adaptive immune characteristics, with HFP birds having lower IgM NAb but higher IgG NAb levels, specific antibody levels, and nitric oxide production by monocytes compared to LFP birds [97]. These findings suggest that HFP and LFP birds differ in immune responsiveness and provide further support to a relationship between the immune system and feather pecking. Yet, these relationships could be the result of genes that are simultaneously involved in the immune system and in feather pecking, as also indicated by previous studies [89,98].

It remains to be elucidated whether these relationships between the immune system and feather pecking are causal. Preliminary findings show that the immune system may play a role in feather pecking. Birds that received an immune challenge at a young age showed more feather damage at an adult age [99], suggesting that activation of the specific immune response at a young age may stimulate birds to feather peck. Following this rationale, it can be considered that a health issue in a flock, such as infection implying immune system activation, may increase the risk of feather pecking in the future. More research is needed on this topic.

#### 3.1.2. Other Impacts of Health on Damaging Behaviour

The health and integument status of laying hens are closely related. Plumage presence, persistence, and distribution on the body can be indicative of the nutritional status, health, and behaviour of the birds [25,100]. Close inspection of growing feathers can also provide information about physiological and systemic infectious issues while the feathers are formed.

Other dimensions of health, such as parasitic infestation, may affect the development of damaging behaviour in laying hens. Parasitic infestation, for example, with *Ascaridia galli*, can decrease health, performance production, and plumage coverage in layer flocks [101]. In this study, parasitic infestation was significantly associated with plumage damage, while treated animals showed better plumage conditions. The authors claim that lower worm burdens were associated with improved plumage condition, possibly through reduced parasite-induced stress, without providing a precise explanation of the mechanism. These results are consistent with the previous hypothesis of this review, where immune stimulation might trigger feather pecking.

Concerning external parasites, red mite (*Dermanyssus gallinae*) infestation can cause anaemia, while the presence of red mites can also lead to itching, disturbing the flock, and possibly acting as a trigger for injurious pecking [100]. The poultry red mite is the most common ectoparasite on laying hen farms worldwide, causing considerable economic losses and reduced hen health and welfare. Even in moderate numbers, they can cause considerable stress, agitation, and severe feather pecking in hens. As an example, it was shown in a study undertaken in 47 Belgian aviaries that the plumage condition of the flock is better on farms with no red mite infestations [25]. Temple et al. [102], in an experiment where infested layers were treated with fluralaner (Exzolt^®^), showed improvements in behavioural variables (less preening, head scratching, head checking, severe feather pecking, and aggressive behaviour), physiological biomarkers, and health parameters following the elimination of red mites on a commercial farm. These results indicate that infestations can reduce hen welfare. The severity of feather pecking associated with red mite infestation may increase in non-beak-trimmed flocks.

Other mites, such as the northern fowl mite (*Ornithonyssus sylviarum*), are also key pest species for caged laying hens. Jacobs et al. [103] showed that mite-infested hens had increased nocturnal activity, including preening, as well as fragmentation of behavioural activities together with decreased dozing, indicating disturbed resting behaviour and suggesting a reduction in the welfare of hens infested by these mites.

Plumage and integument damage can also result from clinical diseases, such as diarrhoea or nutrient deficiency. Hens perform more feather pecking when diets contain mineral, protein, or amino acid (methionine, arginine) levels below recommended levels [104]. Systemic bacterial infections such as *Erysipelas* can be associated with poor feather coverage and skin damage [100].

These findings indicate that health issues may stimulate damaging behaviour, but more research is needed to explain the mechanisms involved and to identify prevention strategies. The following chapter explores the consequences of damaging behaviour on laying hen health outcomes.

### 3.2. Impact of Damaging Behaviours on Health

When discussing the effects of damaging behaviour on the physical and mental health of laying hens, we are primarily referring to the “victim”, i.e., “the recipient”. First of all, the feather-pecking activity may degrade feather cover in recipients, which may interfere with the bird’s body heat regulation, and hens that have lost parts of their plumage are extremely susceptible to the cold [105]. Chickens are sensitive to touch; their skin contains numerous receptors for temperature, pressure, and pain [106]. In crowded systems, feather loss may give rise to skin damage caused by abrasion from the environment and flock mates [57]. Additionally, skin damage can trigger cannibalism [107], often resulting in the mortality of recipients. It has been shown that the victims of cannibalism have lower body weight than feather peckers [108,109]. Furthermore, feather damage may impact the structural cohesiveness of the feathers and lower the aerodynamic capacity of the wings [110,111], making them less efficient in helping to maintain balance [112], which can be problematic when using perches and navigating through a complex 3D aviary environment.

Even feather removal is a strong stressor for a bird; during feather pecking, the bird being pecked often shows crouching immobility with no outward sign of pain. Gentle [113] explained this immobility as learned helplessness, which develops when an animal experiences traumatic events that are aversive and that continue to happen independently of any attempts by the animal to reduce or eliminate them. Studies have shown that during initial feather removal, the birds become agitated, with wing flapping and/or vocalisation and increased heart rate, blood pressure, and EEG arousal as clear signs of pain. Over time, the continued removal of feathers does not produce an exaggerated escape response but an immobile “helplessness” state. During this period of immobility, the EEG of the victim shows activity similar to that seen in sleep or catatonic states, such as tonic immobility. Basically, this is an anti-predator strategy following capture to prevent further damage produced by struggling and to allow escape should the occasion arise. This strategy is, however, counterproductive in production systems where hens have no possibility to escape and are, in effect, making themselves available to be pecked [113]. This type of learned helplessness or anticipation of the negative event may lead to the appearance of negative emotions in hens related to fear and anxiety [106].

Tahamtani et al. [109] suggest that feather peckers and victims experienced similar levels of negative experiences during rearing, causing stress and developmental instability, leading to either pecker or victim status. For example, it is considered that fearfulness, proactive coping, or hyperactivity may predispose chickens to develop severe feather pecking. In the study by Kops et al. [114], the severe feather-pecking problem was discussed because of the lack of monoamines (serotonin and dopamine) in certain brain areas, which affects both emotional perception and behavioural output. Due to neurochemical deficits early in life, high-feather-pecking-line chickens are prone to increased general behavioural activity. In turn, this hyperactivity seems to be a clear risk factor for the development of feather pecking.

To conclude, damaging behaviour leads to denuded overall plumage, with an increased risk of poor thermoregulation, skin damage, and possibly wounds with an increased risk of infection (infection of the skin and tissues and peritonitis). These effects act negatively on hen health and welfare and possibly lead to increased mortality [18,27,50,60].

Consequently, there is a clear need to monitor laying hen health and welfare in order to ensure early detection of damaging behaviour and/or health issues and to use corrective measures. Most modern poultry husbandry systems house thousands of animals in a single barn, leading to challenges in the assessment of individual animals. The next chapter will summarise current knowledge on monitoring systems allowing early detection of damaging behaviour and health issues in order to prevent their spread.

## 4. Systems for Early Monitoring of Animals in Modern Housing Facilities in Order to Limit Occurrence and Spread of Both Health Disorders and Damaging Behaviour

Monitoring of damaging behaviours and health of laying hens can be performed through monitoring of the animals themselves, e.g., behaviour or body condition, or through monitoring of resources, including feed or water consumption and egg production.

### 4.1. Monitoring Tools Based on Direct Observation

In order to identify the risk of compromised health and damaging behaviours at an early stage, it is essential to develop effective and efficient quantitative assessment methods that can easily be applied on commercial farms. Several methods have been developed in order to assess animal welfare in animal husbandry, consisting of the collection of different animal health or welfare parameters from a sample of birds.

The Welfare Quality^®^ [115] method proposes an overall assessment of laying hen welfare on the farm and at the slaughterhouse. Although the evaluation is extensive, the application of the protocol in the livestock requires several hours and needs to be performed by trained assessors. In addition, part of the assessment is conducted at the slaughterhouse and consists of collecting data on indicators that are known to be related to the health and living conditions encountered by the animal on the farm, during transport, or at the slaughterhouse before being killed [116]. The disadvantage is that these are post-mortem observations, which do not allow for corrective actions to be taken on the animals or on the management of the farm, if necessary.

To assess feather damage, numeric rating scales for scoring schemes have been developed and employed in past studies. Current scoring methods [115,117,118,119] differ in the details they record, the type of feathers assessed, the number of body areas assessed, and whether or not birds are captured and handled during the assessment. For instance, Decina et al. [120] compared two feather scoring systems [112,119] based on user-friendliness and reliability [120]. The AssureWel scoring system is the easiest to use and achieves the most consistent outcomes among scorers for the back area of the body. The LayWel system does not provide descriptive definitions of the scores but rather provides photographs as a reference (1–4 scoring scale), while the AssureWel [121] system provides both definitions of scores (0–2 scoring scale) and photographs. AssureWel proposes an overall method of assessment based, for instance, on feather loss, bird cleanliness, observation of antagonistic behaviours, and flightiness.

Animals can be stressed by protocols that require them to be handled for close examination of their physical condition, which may affect some results [122]. To avoid this source of stress, a monitoring approach can be used based on line transects [122,123,124,125,126,127,128]. The transect method assesses the frequency at which animals show clear signs of impaired welfare by noting their incidence while walking along predefined paths or transects that are established among the corridors delimited by drinkers and feeder lines. A new method adapted to aviary has been developed by Vasdal et al. [128], where all the birds observed with feather loss are noted, including those on the littered floor, in the width of the space under the aviary structure, and on each tier of the structure. The scores are standardised by the estimated number of birds in the surveyed area, thus enabling comparisons of the prevalence of various welfare issues between flocks under different husbandry conditions. Several tools have recently been proposed on this principle, sometimes with the development of a smartphone app for easy collection of data and poultry welfare self-assessment by farmers, such as EBENE^®^ for broilers and hens [129], or i-Watchturkey and i-Watchbroiler for turkeys and broilers [130]. These methods allow for shorter durations of welfare assessments. They offer producers multiple possibilities to conduct quantitative flock assessment and apply the necessary corrective actions, and multiple possibilities for the industry in the area of digitalisation and to make informed data-based decisions along the production chain.

In general, the simplicity and time efficiency of the methods are critical aspects to encourage the adoption of the protocol by farmers. The Hennovation or Featherwel projects propose recommendations to improve health and welfare and, to some extent, the consequences of damaging behaviours. For instance, Featherwel enables farmers to regularly monitor the flock via frequent inspections, observing bird behaviour and performing feather scoring to identify injurious pecking early on and to help in the implementation of strategies before the problem becomes more serious.

These methods have the advantage of relying on bird observation and reinforcing the relationships between the farmer and the layers. However, they are time-consuming and, therefore, cannot be run in a continuous manner, allowing only episodic assessment. Other automatic methods allowing continuous assessment of bird health and welfare are detailed below.

### 4.2. Monitoring Tools Based on Precision Livestock Farming

A wide range of sensor technologies can be used to monitor and control damaging behaviour while also minimising consequences on animal health and welfare [131]. Precision livestock farming (PLF) enables real-time and continuous monitoring and management of livestock using modern sensor technologies [132]. In this way, a problem can be identified and diagnosed during the lifetime of the animal so that appropriate corrective measures can be taken immediately if alert criteria are exceeded and before the problem worsens. PLF covers the field of sensors that carry out measurements on animals or in their environment and information and communication technologies that are used to store and transfer data.

#### 4.2.1. Group Monitoring

According to Rowe et al. [133], most PLF strategies use image analysis to measure welfare in poultry farming (42% out of 264 publications). This is because surveillance camera systems combined with image processing techniques are inexpensive ways of providing objective measures of poultry behaviour without having to enter the barn, which involves behavioural changes in the animals. The most common video analysis method is based on counting and identifying small squares (pixels) that turn on and off for a given period of time. Specifically, these methods analyse the variation in brightness or intensity of pixels (on or off) per area of an image, both in time and space. The general idea of these methods is based on the relationship between the number of pixels that turn on and off and the activity of animals in a given unit of area. This method uses cameras to take pictures and analyse the flow. The algorithm then automatically and continuously generates four aggregated statistical values over 15 min sequences (mean, variance, skewness, and flattening) [134]. This method can quantitatively assess variations in the activity of the poultry flock (at the group level) but does not directly account for the welfare of the animals. To do this, individualised monitoring is necessary. For instance, Lee et al. [135] have used optical flow measures as indicators of bird movement, thanks to measures of disturbance using hidden Markov models. Based on these disturbance measures and age-related variables, the authors were able to predict the levels of severe feather damage in flocks in future weeks.

The use of microphones appears to be less widespread in poultry farming (14% of publications [133]). However, sound signals play an important role in animal communication, and some signals may reflect the welfare and health status of the animals. They are used to warn other animals or to communicate with each other, for example, to maintain contact or attract other animals [136]. The *Gallus gallus* species expresses at least 24 different calls to communicate. Chicks between 2 and 3 days old have a repertoire of different vocalisations, from distress calls to pleasure trills and fear trills [137,138]. Certain vocalisations can easily be seen as indicators of animal welfare status [139]. The finer characterisation of vocalisations enables the measurement of welfare indicators reflecting the emotional state of the birds (e.g., warning calls, coughing). The study of these acoustic indicators has made it possible to highlight in recent work an inverse relationship between the live weight of the animals and the peak frequency of their vocalisations. This could enable farmers to identify deterioration in poultry performance early or to predict the weight of animals at slaughter [140,141]. More specifically, acoustic studies are interesting for detecting stress or panic states or abnormal noise on the farm. For example, teams of researchers have focused on identifying rales, characteristic symptoms of respiratory infections in poultry [142,143]. A recent study has developed, under experimental conditions, an algorithm for detecting sneezing in groups of 15 to 36 broilers, with an accuracy of 88% and sensitivity of 67% [144]. Today, the digital processing of sound signals allows various digital descriptions and statistical examinations of the animals’ vocalisations [136]. However, the extraction of sufficient, high-quality signals from animals remains a problem, and well-adapted procedures are required, including noise suppression to remove parasitic noises, such as ventilation noise. Like in research in imaging, artificial intelligence (AI) techniques are being developed for sound signal processing.

#### 4.2.2. Individual Monitoring

In recent years, it has become increasingly possible to monitor individual animals, even within large groups, such as in non-cage systems [145]. A very successful example is that of the dairy cow sector, where it has become standard on many farms for every cow in the herd to be equipped with a sensor and for performance and health to be tracked continuously and fully automatically, with clear positive effects on health, welfare, and production. In the poultry industry, a range of PLF applications has been explored to track individual animals and their health. Tracking allows the recording of information at the individual level, such as the location of the animal, the distance travelled, or the speed of movement. Some solutions require the animals to be individually tagged in order to be tracked. For instance, Banerjee et al. [146] attached wireless sensors to laying hens to monitor their individual activity [146]. Zaninelli et al. [147] used radio-frequency identification (RFID) transponders that were injected into the hen’s feet to collect data on individual behaviour and laying performance (the transponder was injected into the interdigital portion of each hen’s right foot). Injecting the sensor technology into the animal reduces the impact of wearing a sensor, although studies have shown that this impact is minimal and that hens habituate quickly to wearing them [148,149]. Active, ultra-wideband RFID systems have proven to be promising to monitor the location and activity of individual birds, especially when combined with accelerometers, which can provide information on very specific behaviour such as feather pecking [150].

Different tagging technologies can be used, but in some cases, tagging is not suitable for young chicks [151]. So far, these individual monitoring systems are only suitable for experimental studies, and those for laying hens have been tested on rather small samples of birds in a research setting. They are not yet commercially available. Reasons for this are mainly technical, including interference of sensors with the environment, overlap of detection zones in the layer house, and short battery life of the sensors [145]. Another reason could be the cost of equipping every single bird with a sensor. However, the price of this type of technology has been dropping significantly in the last few decades as more and more researchers and producers are exploring the use of sensor technology for livestock production [152]. In the specific context of the poultry sector, individual tagging technologies can be used in a more explorative way, for instance, to assess the different housing systems and their impact on production, health, and welfare. However, it is challenging to develop this system in the field due to the very high number of animals to equip (duration for attaching and removing devices from each individual before slaughtering, data treatment, etc.).

Tagless tracking solutions are also being developed with the use of video. Several steps are required. The first stage of tracking is the detection of individuals in each frame of the video. For the detection of individuals, segmentation is a classic solution that works with low animal density but is sensitive to illumination because they are based on the intensity (brightness) of the pixels. Moreover, even though one can determine with these methods whether a pixel belongs to a chicken or to the ground, it is still complicated to determine which chicken it belongs to when two animals are close to each other.

Faced with the limitations of classic segmentation methods, for example, in the case of higher animal densities, researchers now use AI. Supervised AI allows learning by the machine by showing it thousands of labelled and categorised examples. In this way, the machine becomes capable of correctly classifying most of the images it is shown [153]. A database of characterised images is needed for learning the model, but it is most useful when the model is deemed functional. AI-based detection is much more robust and faster than conventional methods. The major limitation of AI is that more powerful and high-performance machines are needed to allow for great numbers and sometimes more complex calculations. The next step after detection is the tracking of individuals. This does not specifically require the use of AI; classic methods can be used. Recently, a team has started to carry out tracking without marking a small number of laying hens (5) in controlled conditions [154].

In summary, the recent shift to more non-cage production systems in the European Union has created the need for new ways of monitoring and managing the health and welfare of individual laying hens. At the current rate at which technology is evolving and sensor prices are dropping, a sensor for each individual laying hen is not some far-off frontier. Individual monitoring of laying hens will enable farmers to keep track of the health status and behaviour of their birds and to anticipate the spread of damaging behaviour or infections, for example, by removing birds from the flock that are showing pecking damage or symptoms of an infection, indicated by reduced activity or feeding behaviour. Finally, the data from sensors can be used to optimise breeding programmes and to breed out traits such as feather pecking in the long term.

## 5. Conclusions

This review shows the close relationships between damaging behaviours and health in modern husbandry systems for laying hens, which increasingly house the animals in cage-free groups of thousands of birds. These new housing conditions will offer birds more freedom to fulfil their behavioural priorities and, consequently, will reinforce interactions between animals. In case of suboptimal rearing and/or housing and management conditions, damaging behaviour or infectious diseases will be likely to spread to the whole flock. Additionally, health issues and, therefore, stimulation of the immune system may, in certain situations, lead to the development of damaging behaviours, which in turn may result in impaired body condition, leading to further health and welfare issues. This highlights the need to monitor both behaviour and health of laying hens in order to intervene as quickly as possible to preserve the health and welfare of animals, as well as farmer income and work satisfaction.

## Data Availability

Not applicable.

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
