# Peer review of "The Relationships between Damaging Behaviours and Health in Laying Hens"

_animals, 2022, doi:10.3390/ani12080986_

Round 1

Reviewer 1 Report

Dear authors,

thank you for this important review about the relationships between damaging behaviours and health in laying hens.

Find the revision note attached

Author Response

Line 9, 13, 18: You are inconsistent in giving semicolons after the email addresses This has been corrected

Line 98: In how far is the information connected to FP? Maybe delete the sentence? Sentence has been deleted

Line 152: The paragraph introduces with ‘unwanted behaviors’ and mentions aggressive pecking in this course. Since aggressive pecking is part of the normal or natural behavior and is necessary to keep the hierarchy, I wonder if we do not want this behavior in laying hens? Thank you, we agree that aggressive pecking is a natural/normal behavior, however it is unwanted when it becomes too pregnant, too frequent, out of the “normal” hierarchy establishment. This has been precise.

Lines 433-436: Are you sure that visual assessment without handling leads to the similar results like ‘handling scoring’? I would formulate this more carefully. Bright et al. [124] found a high correlation between both assessment methods. Have you proofread the linearity of this correlation? Unfortunately, I have only access to the abstract. I’m not sure in how far the standard deviation of 1.5 means a deviation of 1.5 on the one side and 1.5 to the other side, that would be a high difference when you have a 5-point scale. Kjaer et al. [129] wrote in their result section: “In general, kappa and weighted kappa values between methods and teams for the total plumage score ranged from 0.25 to 0.34 and all were significantly larger than zero (Table 2). PABAK values for total scores were higher (0.35 to 0.55)”. These reliabilities are not acceptable (e.g. Landis, J. Richard; Koch, Gary G. (1977): The Measurement of Observer Agreement for Categorical Data. In: Biometrics 33 (1), S. 159. DOI: 10.2307/2529310 and Gunnarsson, S.; Algers, B.; Svedberg, J. (2000): Description and evaluation of a scoring system of clinical health in laying hens. In: Gunnarsson, S.: Laying hens in loose housing systems; clinical, ethological and epidemiological aspects. PhD-Dissertation Swedish University of Agricultural Sciences, Uppsala). Of course, in their table much higher and better PABAK values were mentioned, but I’m not sure how these different results should be interpreted. This is very interesting, thank you. This needs indeed deeper investigations. That is why we prefer to remove this part in the manuscript.

Line 545-546: I would prefer to restructure a bit this paragraph. What advantages would individual tracking have over PLF systems that are group-based? Who should attach the transmitters to the animals and, above all, who removes the transmitters before depopulation? Or should they end up in the slaughterhouse? Resource consumption? Who evaluates all the data? How can interactions between two animals be recorded with it? I think before recommending individual tracking, some more thought would need to be put into it. Or in the section there should be a better distinction between scientific studies and assessment for the praxis. We have now precised that individual tagging technologies can only be used in a explorative way, and remind all these limits. Thank you.

Reviewer 2 Report

In general

This review represents a valuable contribution to the problem of feather pecking and health. In some cases, the literature needs a more critical view. Publications which are not in line with the given text should not be neclected (see details below).   

The authors quote a large number of studies on the immune system and feather pecking. Is there any evidence that commercial flocks showing high feather pecking are more susceptible to infectious diseases than those showing low feather pecking?

In Particular:

Lines 62 – 64: Feather pecking and cannibalism have been reported frequently even before the development of industrial intensive production systems. See Montellano, J. (1901): La Gallina y otras aves de coral. Barcelona: Francisco Sabater. p. 217.

Line 95 feather pecking is painful for the birds being pecked

Line 143 “This suggests that severe feather pecking is not a direct substitute for foraging, but that some individuals have high pecking motivation overall and are, thereby, more prone to develop injurious pecking.”

This is difficult to understand. Does it mean, that birds with a high motivation of general pecking perform feather pecking but no foraging?

Lines 154 – 156: cannibalism usually comprises vent/cloacal pecking and toe pecking

Line 208 – 209: selection for high locomotor activity did not result in an increase of feather pecking. (Krause et al , 2019, doi: 10.1016/j.beproc.2019.103980.)

This is in contrast to the hyperactivity hypothesis.

Line 213 – 217: In contrast,  Borda-Molina et al. (2021) (Life 11(3):235. doi:10.3390/life11030235) did not find a relationship between gut microbiome and feather pecking.

Lines 376 – 378: “This type of learned helplessness or anticipation of the negative event may lead to the appearance of negative emotions in hens – related to fear and anxiety – leading to chronic stress [112].”

Poor feathering is not directly related to egg production and mortality. There exist flocks which are almost denuded but which show maximum egg production and low mortality. (see more refrences by Boegelein et al. (2015)DOI: 10.1399/eps.2015.84. It is well established that chronic stress has a negative impact on both criteria. Hence the assumption that removal of feathers through feather pecking leads to chronic stress is highly speculative.

Line 487 “Lee, et al…” delete comma

Author Response

In general : This review represents a valuable contribution to the problem of feather pecking and health. In some cases, the literature needs a more critical view. Publications which are not in line with the given text should not be neclected (see details below). The authors quote a large number of studies on the immune system and feather pecking. Is there any evidence that commercial flocks showing high feather pecking are more susceptible to infectious diseases than those showing low feather pecking? To our knowledge, no direct relationship between feather pecking and susceptibility to infectious disease has been studied and proved.

In Particular:

Lines 62 – 64: Feather pecking and cannibalism have been reported frequently even before the development of industrial intensive production systems. See Montellano, J. (1901): La Gallina y otras aves de coral. Barcelona: Francisco Sabater. p. 217. => True, modified into : “(…) behavioural disorders such as feather pecking, toe pecking, vent/cloacal pecking and cannibalism can occur.”

Line 95 feather pecking is painful for the birds being pecked. This has been added

Lines 154 – 156: cannibalism usually comprises vent/cloacal pecking and toe pecking – true, modified into different forms of cannibalism among which it can be found vent/cloacal pecking and toe pecking

Line 208 – 209: selection for high locomotor activity did not result in an increase of feather pecking. (Krause et al , 2019, doi: 10.1016/j.beproc.2019.103980.) This is in contrast to the hyperactivity hypothesis. Although high locomotor activity is not specially linked to hyperactivity, we added this interesting reference, thank you.

Line 213 – 217: In contrast, Borda-Molina et al. (2021) (Life 11(3):235. doi:10.3390/life11030235) did not find a relationship between gut microbiome and feather pecking. Thank you, this reference has been added

Lines 376 – 378: “This type of learned helplessness or anticipation of the negative event may lead to the appearance of negative emotions in hens – related to fear and anxiety – leading to chronic stress [112].” Poor feathering is not directly related to egg production and mortality. There exist flocks which are almost denuded but which show maximum egg production and low mortality. (see more refrences by Boegelein et al. (2015)DOI: 10.1399/eps.2015.84. It is well established that chronic stress has a negative impact on both criteria. Hence the assumption that removal of feathers through feather pecking leads to chronic stress is highly speculative. Reference to chronic stress has been removed.

Line 487 “Lee, et al…” delete comma corrected

Reviewer 3 Report

This is an extremely well written review and covers a large section of important information on the topic. References are extensive (159 referenced articles) and of high quality. In some instances references are cited for minimal additive value and you might consider decreasing the number and citing in more depth, but generally a good review.

Author Response

Although the paper addresses a lot of topics, we have revised the number of references as much as possible from 159 to 151.